# Facile Fabrication of a Self-Healing Temperature-Sensitive Sensor Based on Ionogels and Its Application in Detection Human Breath

**DOI:** 10.3390/nano9030343

**Published:** 2019-03-03

**Authors:** Fengxia Wang, Shaohui Zhang, Yunlin Zhang, Qihang Lin, Yun Chen, Dongfang Zhu, Lining Sun, Tao Chen

**Affiliations:** 1Jiangsu Provincial Key Laboratory of Advanced Robotics, Soochow University, Suzhou 215123, China; wangfengxia@suda.edu.cn (F.W.); 20175229052@stu.suda.edu.cn (Y.Z.); 20185229014@stu.suta.edu.cn (Q.L.); lnsun@hit.edu.cn (L.S.); 2Scicence and Technology Department, Shanghai Aerospace Control Technology Institute, Shanghai 201109, China; 13818278790@139.com (Y.C.); dongfang0704@126.com (D.Z.); 3Suzhou Institute of Nano-tech and Nano-bionics, Chinese Academy of Sciences, Suzhou 215123, China; shzhang2016@sinano.ac.cn

**Keywords:** ionogels, self-healing, temperature-sensitive, sensor, human breath

## Abstract

The biocompatible strechable ionogels were prepared by a facile solution-processed method. The ionogels showed outstanding stretchable and self-healing properties. The electrical property could revert to its original state after 4 s. The repaired ionogels could still bear stretching about 150%. Moreover, the ionogels exhibited high sensitivity and wide-detection range to temperature. The temperature-sensitive sensor could detect the human breath frequency and intensity, showing potential application in detecting disease.

## 1. Introduction

Ionic conductors’ materials have attracted attention due to the potential application in solar energy conversion devices and flexible sensors, such as a solar device and pressure sensor [1,2]. Generally, there are two main ionic conductors, including the hydrogels and ionic gels. The hydrogels with electrolytes have shown excellent flexible, transparent, conductive, and biocompatible properties. They have been widely applied in high-performance actuators, highly sensitive pressure sensors, and outstanding performance strain sensors [3,4,5]. Despite the great achievement in the fabrication and the devices of hydrogels, the limited stability of the aqueous electrolytes still restricts their further application as wearable flexible sensors.

Comparably, the ionogels possess the attributes of elastic solid form, excellent flexibility, high thermal stability, high ionic conductivity, nonvolatility, and biocompatibility [6,7,8]. They have shown intensive applications in fuels [9], radical scavenging activity [10], antibacterial activity [11], environmental remediation [12], and flexible electric devices [13,14,15], particularly in the wearable sensor fields [16,17,18]. Recently, great achievements have been obtained in the fabrication of materials and the construction of wearable sensors based on the ionogels [19,20]. It is worth noting that the cross-linked ionogel agents with the no-conductive polymer often show low sensitivity and weak mechanical strength [14,21,22]. By comparison, the ionogels based on polymerizing ionic-liquid (IL) monomers exhibit potential applications in high conductive electrodes and wearable sensors [23,24]. However, most reported wearable sensors often focus on a single performance, such as sensing pressure, stretching, or temperature [24,25]. As we know, most wearable sensors, such as breath sensing sensors, not only need high sensitivity to achieve good measurement accuracy, but need self-healing properties to compensate for their vulnerability to damage under continuous stretching or pressing [26,27]. Thus, the development of an ionogel-based multifunctional sensor with self-healing characteristics is very vital in wearable e-skin electrics, although it remains a considerable challenge.

Herein, the biocompatible stretchable ionogels were fabricated by a facile solution-processed method combined with SiO_2_ particle. The obtained ionogels showed high transparency, high conductivity, excellent stretchability, and self-healing characteristics. When the ionogels were cut into two pieces, their electrical properties could recover to the original state only after 4 s. The repaired ionogels could still bear stretching about 150%. In addition, the ionogels exhibited high sensitivity and wide-detection range to temperature. The temperature-sensitive sensor could detect different human respiration characteristics under different conditions, which showed the application prospects in health monitoring and rehabilitation treatment fields in the future.

## 2. Materials and Methods

### 2.1. Preparation

In total, 12.5 mg ammonium persulfate (APS) and 25 μl silica microsphere were added into 1 mL 1-vinyl-3-ethylimidazolium dicynamide ([VEIm][DCA]) ionic liquid. The mixture was shaken for 20 min by a vortex shaker. The solution was dropped on the channel of the flexible substrate and then polymerized at 80 °C for 2 h. Subsequently, the obtained ionogel was washed by NaOH solution and deionized water. The copper foils were attached on the ionogel’s film surface as electrodes. Then, another fabricated Eco-flex film was put on the surface of the ionogel’s film with copper electrodes to improve the stability. Subsequently, the solution of the Eco-flex was poured on the side of the two Eco-flex films. The obtained covered films with electrodes were put in a 70 °C oven for 20 min, and the flexible sensors based on the ionogels were prepared. In this work, the thin ionogel film was difficult to form due to not being polymerized. For the thick film, the flexibility would decrease significantly. Thus, the thickness of the film was set as about 500 μm.

### 2.2. Characterization

For the morphology and element analysis, the ionogels were investigated by scanning electron microscope (SEM, Quanta FEG250, FEI company, Hillsboro, OR, USA). Raman spectra of the ionogels were recorded on a Horiba LABRAM HR Raman spectrometer (Horriba-JY company, Kyoto, Japan) with an excitation wavelength of 633 nm and spectral range of 100–3500cm^−1^. Fourier transform infrared spectra (FT-IR) were measured by a Nicolet iN10 spectrometer (Nicolet Instrument Corporation, Madison, WI, USA) with spectral range of 600–4000cm^−1^, test mode of transmission, liquid nitrogen retention time of 700 mL/18 h, slide of gold piece, and sample thickness of 30–40 μm. Thermo gravimetric (TG) and Differential Scanning Calorimetry (DSC) were recorded by the Thermogravimetric/differential thermal analyzer (TG/DTA6200, Seiko Instruments Inc., Nagano, Japan). The temperature ranged from room temperature to 500 °C with heating rate of 10 °C/min and nitrogen was used as protective gas. Conductive performances under different conditions were recorded by the Keithley semiconductor meter (4200) (Tektronix, Beaverton, OR, USA) at room temperature. Mechanical and stretchable properties were performed by Mark-10 (Mark-10 Corporation, Copiague, NY, USA).

## 3. Results and Discussions

The fabricated steppes of the ionogels were illustrated, as shown in Figure 1. It was obvious that the fabricated process was very facile and low-cost, which was suitable for combining with the printing method, resulting in large-scale production. As shown in the inset of Figure 1a, the ionogels had micro-structure morphology. The micro-structure size was about 1 μm, which matched the diameter of the inserted SiO_2_ particle. Moreover, the ionogels were fully flexible and could be bent and twisted randomly (Figure 1b,c). The ionogels were very stretchable and could be strained about 30 times (Figure 1d), which was significantly better than the reported literature [16,24]. The conductivity of the ionogels almost linearly decreased with the stretching length. In addition, the ionogels showed high transparency and biocompatibility (Figure 1e). These above excellent characteristics proved their potential application in wearable sensors.

To further investigate the composition and the properties of the ionogels, spectroscopic analyses were adopted using Fourier transform infrared (FT-IR), Raman spectroscopy, and the Energy Dispersive X-ray Detector (EDX). As shown in Figure 2a, the peaks from the (ionic liquids) ILs and the (ammonium persulfate) APS all disappeared, while two new peaks were observed in the FT-IR after reaction. As shown in Raman spectra, additional peaks were observed, and the obvious peaks came from the ILs, while the main peaks from the APS disappeared. These results demonstrated that the ionogels were formed consisting of ILs. In addition, the EDX element analysis spectrum exhibited only C, N, and O peaks without the S peak from the APS, which was further confirmed the above results (Figure 2c). It was noted that the ionogels also obtained good thermo stability. There was a two-step weight loss process when the temperature rose from room temperature to 450 °C using the 10 °C /min rate (Figure 2d), similar to the reported literature [24]. The first changeable range of the ionogel film began at 50 °C and almost finished at 190 °C, where the heat loss of the ionogels was about 8% weight loss only. It was obvious that the heat loss increased dramatically with the further increase of the temperature, which were almost 50% at about 450 °C. In addition, the melting temperature (Tm) of the ionogels was ~266 °C, corresponding to the sharp endothermic peaks in the in the DSC thermogram of the ionogels (Figure 2d).

Besides the flexibility and stretchability, the self-healing capability of the ionogels was also investigated. The ionogel film was first cut into two pieces, and then the two pieces of the ionogel films were put together in order to heal. After 24 hours, the damage site of the film almost completely disappeared, indicating the excellent healing property of the ionogels (Figure 3a). As expected, the conductivity of the repaired ionogels would recover to its original state immediately (Figure 3b). Moreover, the self-healed film could still bear stretching again to about 150% (Figure 3c), further demonstrating the excellent self-healing and stretchable properties. Herein, the stretchability of the ionogels was assessed by the (*L* − *L*_0_)/*L*_0_, where *L*_0_ and *L* were the length of the ionogels without stretching and after stretching, respectively. To further prove the excellent and fast-response self-healing properties, a circuit was constructed and shown in Figure 3d. The light-emitting bulb was on when the circuit was connected to the ionogel film, demonstrating the highly conductive characteristics of the ionogel. When the film was cut into two separate pieces, the light-emitting bulb was off. It was noted that the light-emitting bulb was on again after the two cut pieces of the ionogels film were put together for around 5 second. Moreover, the self-healed circuit based on the ionogels could work well. The above results further demonstrated the excellent self-healing and highly conductive properties.

The temperature-current characteristics of the ionogels were also measured and presented in Figure 4a and Figure 4b. It can be seen that the current increased with the increase of the temperature, which was consistent with reported temperature sensors based on ionogels [28]. The effects of the temperature to the current was attributed to the migration rate of ionic mobility. As we know, ionic mobility was higher at high temperatures and resulted in higher conductivity. Furthermore, the relative current change almost increased linearly with the increase of the temperature from 30 °C to 45 °C (Figure 4c). Thus, the temperature coefficient of current parameter could be calculated by *α* = (Δ*I*/*I*_0_)/Δ*T*, where *I*_0_ is the initial current at room temperature, Δ*I* is the change of current (*I* − *I*_0_), and Δ*T* is the change in temperature (*T* − *T*_0_). The *α* was estimated to be 0.03 °C^−1^. In addition, the sensor showed a wide responding range from 28 °C to 100 °C (Figure 4d). The high sensitivity and the well linear change covering the human temperature demonstrated potential application in temperature-mechanism-based wearable sensors.

Due to the wide detection range and high sensitivity to temperature, the ionogels were promising for detecting health signals from the temperature variation. A wearable nose based on the ionogels was constructed (Figure 5a). The current showed two distinct states, a “low” current state (*I*_0_) under inhaling and “high” current state (*I*) under exhaling. The relative current change ((*I* − *I*_0_)/*I*_0_) was dependent on the breath frequency and intensity (Figure 5b). When taking a deep breath, the relative current change and the response time would become greater compared with a normal breath, while they would become lesser when taking a fast breath (Figure 5c–f). As we know, when exhaling, the high-temperature gas would reach the surface of the sensor, resulting in the increase of the conductivity. When exhaling stopped, the temperature of gas around the sensor would decrease to the room temperature, thus the conductivity would return to the original value. These results indicated the potential application in detecting disease.

## 4. Conclusions

Biocompatible stretchable ionogels were successfully fabricated by the in situ polymerization method, which showed high transparency and high conductivity. Moreover, the ionogels possessed outstanding self-healing properties. The electrical property reverted to its original state only after 4 s and the self-healed film could be stretched up to 150%, which was an important characteristic to compensate for the vulnerability to damage under continuous stretching or pressing of flexible sensors. Importantly, the ionogels exhibited high sensitivity and wide-detection range to temperature. The temperature-sensitive sensor could detect human breathing characteristic signals, showing potential application in detecting disease.

## Figures and Tables

**Figure 1 nanomaterials-09-00343-f001:**
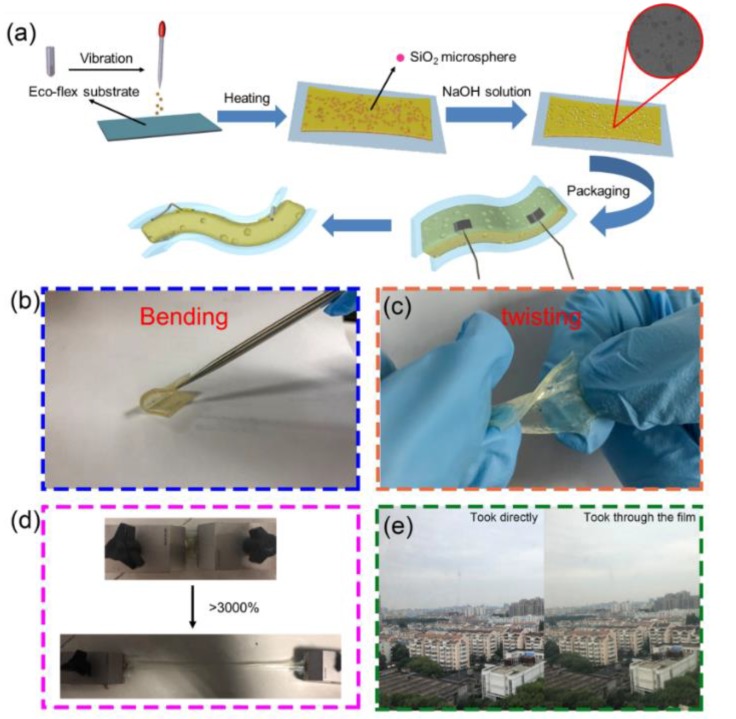
(**a**). The schematic illustration of the fabrication process of the ionogels. (**b**) The photograph of ionogels bending. (**c**) The photograph of ionogels twisting. (**d**) The photograph of ionogels stretching. (**e**) The photograph of ionogel film using the laboratory outside as the background.

**Figure 2 nanomaterials-09-00343-f002:**
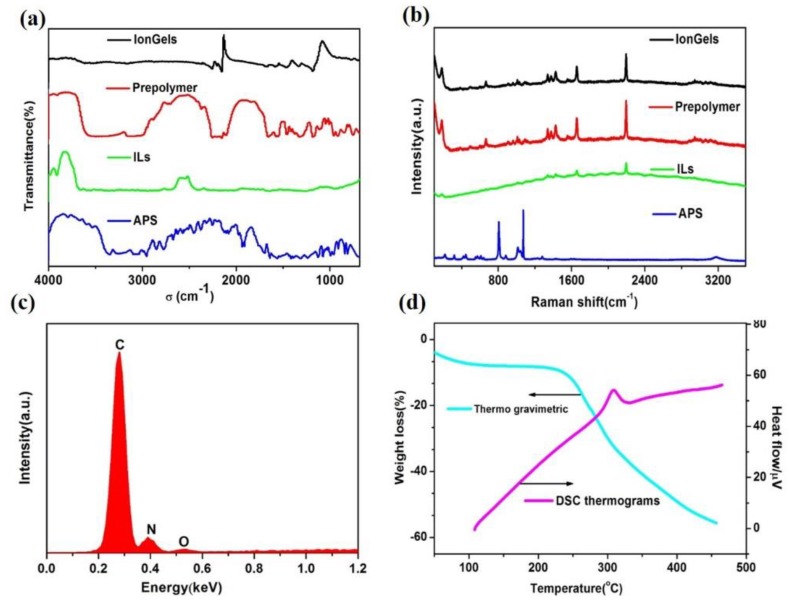
(**a**) Fourier transform infrared spectra. (**b**) Raman spectra. (**c**) Energy Dispersive X-ray spectrum. (**d**) Thermo gravimetric and Differential scanning calorimetry thermo-gram curves of the ionogels.

**Figure 3 nanomaterials-09-00343-f003:**
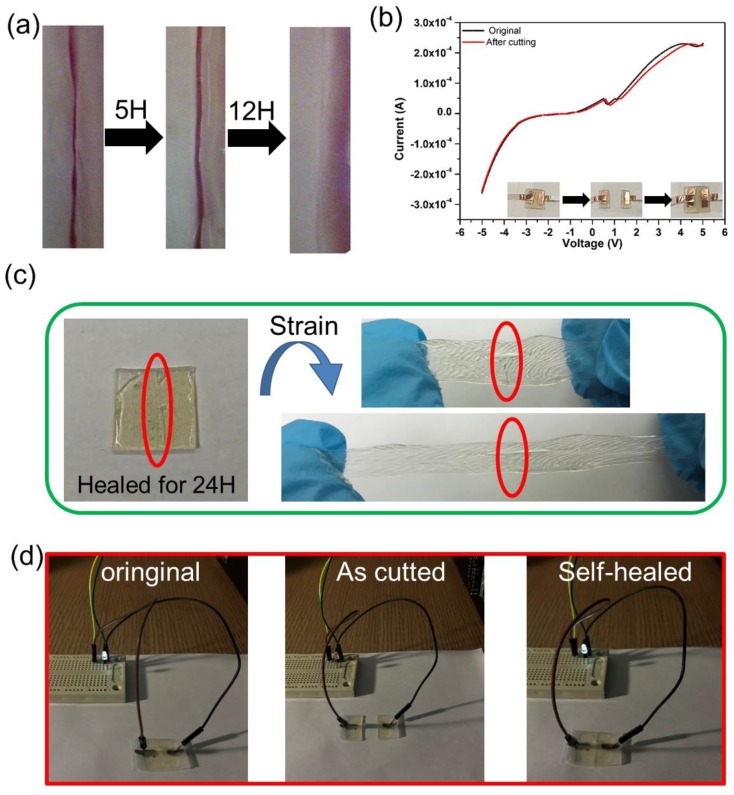
(**a**) The morphology and (**b**) the current-voltage curves of the ionogel film before and after self-healing. (**c**) The photograph of the stretching of the repaired ionogel film. (**d**) The circuits based on the ionogels, cut ionogels, and self-healed ionogels.

**Figure 4 nanomaterials-09-00343-f004:**
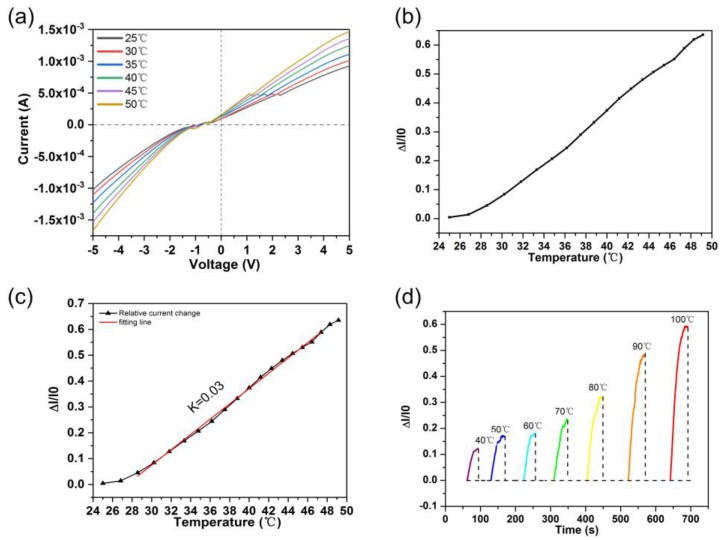
(**a**) Current-voltage curves at various temperatures. (**b**) Relative current change with temperature rising from 25 °C to 50 °C. (**c**) The fitting line of the relative current change with well linear change from 28 °C to 48 °C. (**d**) Current change amplitude under different temperatures.

**Figure 5 nanomaterials-09-00343-f005:**
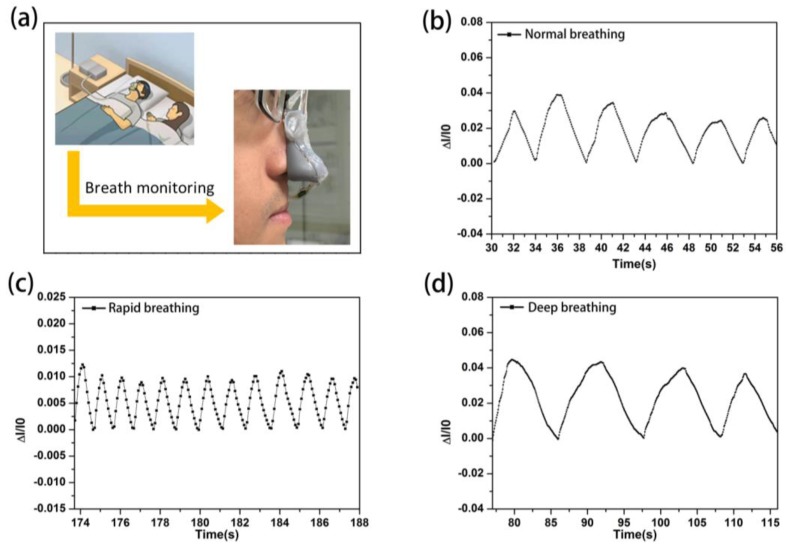
(**a**) Ionogel-based sensor. (**b**) Relative current changes of common respiration. (**c**) Relative current changes of rapid breathing. (**d**) Relative current changes of deep respiration.

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
