# Peer review of "Facile Fabrication of a Self-Healing Temperature-Sensitive Sensor Based on Ionogels and Its Application in Detection Human Breath"

_nanomaterials, 2019, doi:10.3390/nano9030343_

Round 1

Reviewer 1 Report

This manuscript describe a biocompatible strechable iongels with outstanding self-healing properties used for wearable temperature sensor. I think it can be published with the following information provided.

There is no information about how the conductivity change with stretching process;

How the humidity affect on the sensing when detecting human breath;

How about the iongel thickness affect on the performance, like transparency, conductivity and sensitivity.

Reviewer 2 Report

The paper deals with the fabrication of self-healing sensitive sensor based on ionogels. Although, the manuscript could be of a certain interest for people working in the field, it is in a preliminary form and it could be suitable of publication only after major revision.

Some suggestions are following reported:

1)In the Introduction, the AA state that ionogels show poor stretchability and weak self-healing property. The referee does not agree with the above statement as many papers have been published about ionogels and report the self-healing ability of these materials (see for example ACS Applied Materials & Interfaces 2018,10, 5871; Journal of Colloid and Interface Science 2017,487, 130-140; ACS Nano 2018,12(2), 1296-1305; Chemistry – A European Journal 2016,22(32), 11269-11282; Chemistry – A European Journal 2017,23(64), 16297-16311;Green Chem. 2018, 20, 4260; Journal of Colloid and Interface Science 2018,517, 182-193.)

2) The Introduction is quite general and does not allow having a general idea about the topic of the manuscript and the aim of experimental work. For the above reasons, it should be re-written deepening the above aspects;

3) Interestingly, DSC and TGA analysis were reported, but details of experimental methodologies are not reported: They should be inserted. On the other hand, also methodology to obtain Raman and IR spectra must be reported;

4) On page 3, line 95, the AA state that ionogels are biocompatible, but this referee does not understand what is the experimental result supporting the above assertion;

5) The Aa must also explain how they assess the stretchability of the ionogel;

6) The manuscript must be carefully reread as it contains many typos and English mistakes.

Round 2

Reviewer 1 Report

The author answer questions very well and this manuscript can be accepted in the present form. Thanks.